# Flexibility in PAM recognition expands DNA targeting in xCas9

**Kazi A Hossain[1,2], Lukasz Nierzwicki[1], Modesto Orozco[3,4], Jacek Czub[2,5], Giulia Palermo[1,6]***

[1]Department of Bioengineering , University of California Riverside, Riverside, United States; [2]Department of Physical Chemistry, Gdańsk University of Technology, Gdańsk, Poland; [3]Institute for Research in Biomedicine (IRB Barcelona), The Barcelona Institute of Science and Technology, Barcelona, Spain; [4]Departament de Bioquímica i Biomedicina, Facultat de Biologia, Universitat de Barcelona, Barcelona, Spain; [5]BioTechMed Center, Gdańsk University of Technology, Gdańsk, Poland; [6]Department of Chemistry, University of California Riverside, Riverside, United States

## eLife Assessment

This manuscript describes a **fundamental** investigation of the functioning of Cas9 and in particular on how variant xCas9 expands DNA targeting ability by an increase-flexibility mechanism. The authors provide **compelling** evidence to support their mechanistic models and the relevance of flexibility and entropy in recognition. This work can be of interest to a broad community of structural biophysicists, computational biologists, chemists, and biochemists.

***For correspondence:**
giulia.palermo@ucr.edu

**Competing interest:** The authors declare that no competing interests exist.

## Abstract

xCas9 is an evolved variant of the CRISPR-Cas9 genome editing system, engineered to improve specificity and reduce undesired off-target effects. How xCas9 expands the DNA targeting capability of Cas9 by recognising a series of alternative protospacer adjacent motif (PAM) sequences while ignoring others is unknown. Here, we elucidate the molecular mechanism underlying xCas9's expanded PAM recognition and provide critical insights for expanding DNA targeting. We demonstrate that while wild-type Cas9 enforces stringent guanine selection through the rigidity of its interacting arginine dyad, xCas9 introduces flexibility in R1335, enabling selective recognition of specific PAM sequences. This increased flexibility confers a pronounced entropic preference, which also improves recognition of the canonical TGG PAM. Furthermore, xCas9 enhances DNA binding to alternative PAM sequences during the early evolution cycles, while favouring binding to the canonical PAM in the final evolution cycle. This dual functionality highlights how xCas9 broadens PAM recognition and underscores the importance of fine-tuning the flexibility of the PAM-interacting cleft as a key strategy for expanding the DNA targeting potential of CRISPR-Cas systems. These findings deepen our understanding of DNA recognition in xCas9 and may apply to other CRISPR-Cas systems with similar PAM recognition requirements.

## Introduction

Expanding the targeting scope of genome editing systems towards a broader spectrum of genetic sequences is a major priority to advance the CRISPR-Cas technology (*Pacesa et al., 2024*; *Wang and Doudna, 2023*). xCas9 is an evolved variant of the CRISPR-Cas9 system (*Hu et al., 2018*), engineered to improve specificity and reduce undesired off-target effects compared to the *Streptococcus pyogenes* Cas9 (SpCas9) (*Anzalone et al., 2020*; *Pacesa et al., 2024*). In this system, the endonuclease Cas9 associates with a guide RNA to recognise and cleave any desired DNA sequence enabling

**Figure 1.** xCas9 variant of the *S. pyogenes* Cas9 (SpCas9) protein bound to a guide RNA (grey) and a target DNA (charcoal) including the 5′-AGG-3′ protospacer adjacent motif (PAM) recognition sequence (red) (PDB 6AEB) (*Guo et al., 2019*). xCas9 includes seven amino acid substitutions (blue) with respect to SpCas9. Close-up views of the PAM recognition region for xCas9 bound to AGG (PDB 6AEB, left) (*Guo et al., 2019*) and GAT (PDB 6AEG, right) (*Guo et al., 2019*). The PAM nucleobases (red) and the PAM interacting residues (R1333 and R1335, blue) are shown as sticks. The E1219V mutation is also shown.

facile genome editing (*Jinek et al., 2012*). Site-specific recognition of the target DNA occurs through a short protospacer adjacent motif (PAM) sequence next to the target DNA, enabling precise selection of the desired DNA sequence across the genome.

However, PAM recognition is also a significant bottleneck in fully exploiting the genome editing potential of CRISPR-Cas9 (*Anzalone et al., 2020*; *Pacesa et al., 2024*). In SpCas9, recognition is strictly limited to 5′-NGG-3′ PAM sequences, mediated by the binding to two arginine residues (R1333, R1335) within the PAM-interacting domain. This constraint significantly limits the DNA targeting capability of SpCas9, as the occurrence of NGG sites within a given genome is restricted. To address this inherent limitation, extensive protein engineering and directed evolution led to novel variants of the enzyme. The xCas9 3.7 variant (hereafter referred to as xCas9, *Figure 1*) has emerged as a notable advancement (*Hu et al., 2018*), expanding the PAM targeting capability towards a variety of sequences, including guanine- and adenine-containing PAMs. xCas9 not only demonstrates a significantly expanded PAM compatibility but also improves recognition of the canonical TGG PAM and reduces off-target effects compared to SpCas9 (*Hu et al., 2018*). Despite these advancements, the mechanism by which xCas9 expands DNA targeting capability by recognising a series of PAM sequences while ignoring others remains unknown.

xCas9 was developed through directed evolution introducing seven amino acid substitutions within SpCas9 (*Figure 1*). Notably, the only substitution within the PAM-interacting domain is E1219V and does not directly interact with the PAM sequence. Additionally, structures of xCas9 bound to AAG (PDB 6AEB) and GAT (PDB 6AEG) PAM sequences show no substantial differences in the PAM-interacting domain compared to SpCas9 (*Chen et al., 2019*; *Guo et al., 2019*). Hence, this substitution does not explain xCas9's ability to recognise non-canonical PAM sequences, including those with adenine that typically do not favourably interact with arginine (*Hossain et al., 2023*; *Luscombe et al., 2001*). It is unclear how the E1219V mutation, rather than the R1333/R1335 substitution, facilitates binding to alternative PAMs. Additionally, the impact of other xCas9 mutations scattered across the protein on nucleic acid binding has not been addressed.

Here, we establish the molecular determinants of the expanded DNA targeting capability of xCas9 through extensive molecular simulations, well-tempered metadynamics, and alchemical free energy calculations. We demonstrate that while SpCas9 enforces a strict guanine selection through the rigidity of its arginine dyad, xCas9 modulates the flexibility of R1335 to selectively recognise specific PAM sequences, conferring also a pronounced entropic preference for TGG over SpCas9. We also show that directed evolution improves DNA binding, expanding the DNA targeting capability in the early evolution cycles, while achieving tight DNA binding with the canonical TGG PAM through the

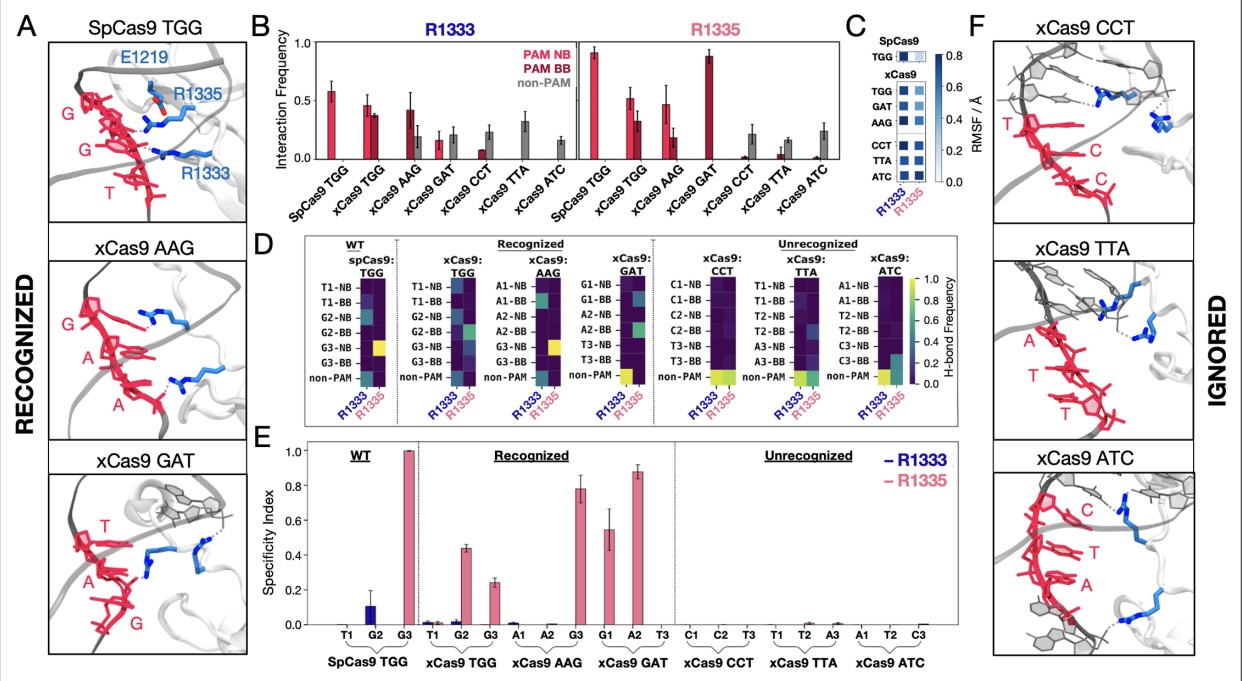

**Figure 2.** Binding of SpCas9 and xCas9 to protospacer adjacent motif (PAM) sequences that are recognised and ignored. (**A**) Binding of SpCas9 and xCas9 to PAM sequences that are recognised (TGG, top; AAG, centre; GAT, bottom). (**B**) Interaction pattern established by R1333 (left) and R1335 (right) with PAM nucleobases (NB), PAM backbone (BB), and non-PAM nucleotides in SpCas9 bound to TGG (i.e. the wilt-type system) and xCas9 bound to PAM sequences that are recognised (TGG, AAG, GAT) and ignored (CCT, TTA, ATC). Interaction frequencies are averaged over ~6 µs of collective ensemble for each system. Errors are computed as standard deviation of the mean over four simulations replicates. (**C**) Root mean square fluctuations (RMSF) of the R1333 and R1335 side chains in SpCas9 bound to its TGG PAM, compared to xCas9 bound to recognised and ignored PAMs. (**D**) Frequencies of hydrogen bond formation between the arginine side chains and the PAM NB, BB, and non-PAM nucleotides (details in the SI). (**E**) Specificity index, representing the frequency of hydrogen bond formation between a given arginine and the PAM nucleotides relative to the frequency of forming hydrogen bonds with non-PAM residues. Data are reported with the standard deviation of the mean over four simulations replicates. (**F**) PAM recognition region in xCas9 bound to PAM sequences that are ignored (CCT, top; TTA, centre; ATC, bottom). *Figure 2—figure supplement 1*.

The online version of this article includes the following figure supplement(s) for figure 2:

**Figure supplement 1.** Frequencies of hydrogen bond formation across four simulation replicates.

last evolution cycle. These findings will facilitate the development of improved Cas9 variants with expanded DNA recognition capabilities.

## Results
### PAM recognition requires specific interactions

To elucidate the selection mechanism, we performed multi-µs molecular dynamics (MD) simulations of xCas9 bound to PAM sequences that are recognised well (TGG, GAT, and AAG) and that are ignored (ATC, TTA, and CCT), and compared to SpCas9 bound to its canonical PAM (TGG). We broadly explored the systems' dynamics and defined the statistical relevance of critical interactions through ~6 µs of sampling for each system (in four replicates, totalling ~42 µs runs).

Since in SpCas9, R1333 and R1335 provide anchoring for the PAM nucleotides (*Figure 2A*; *Anders et al., 2014*), we performed an in-depth statistical analysis of their interactions with the DNA, using both distance and energetic criteria (details in Materials and methods). We analysed the probability for R1333 and R1335 to interact with the PAM nucleobases (PAM NB), the PAM backbone (PAM BB), and non-PAM nucleotides (non-PAM) (*Figure 2B*). In SpCas9, both arginine residues steadily interact with the PAM NB (*Anders et al., 2014*; *Bhattacharya and Satpati, 2024*; *Palermo et al., 2017*), with R1335 exhibiting a higher probability compared to R1333. Analysis of the arginine' flexibility (*Figure 2C*) denotes that R1335 is remarkably constrained in SpCas9, likely a result of its interaction with the PAM NB and its vicinity to E1219, while R1333 is more flexible. Interestingly, in xCas9 bound

to TGG, both R1333 and R1335 switch their interactions between the PAM NB and BB (*Figure 2B*). Similarly, for xCas9 bound to other recognised PAMs (AAG and GAT), the arginine dyad interacts with both PAM NB and BB, with R1335 being more specific than R1333, which also contacts non-PAM nucleotides. Contrarily, for xCas9 bound to ignored PAM sequences (CCT, TTA, and ATC), no significant interactions with the PAM NB or BB were observed, while both arginine primarily interacted with non-PAM nucleotide. This is in line with SpCas9's specificity for NGG PAMs (*Hu et al., 2018*), and with the arginine's preference for guanine (*Hossain et al., 2023*; *Luscombe et al., 2001*). To further detail the interactions established by the arginine dyad and the nucleotides, we analysed the frequency of hydrogen bonds among them (*Figure 2D*). In SpCas9, R1335 secures its interaction with the G3 nucleobase (G3 NB), while R1333 interacts with G2 NB and contacts non-PAM nucleotides. In xCas9 bound to TGG, the arginine dyad maintains its interactions with PAM. Here, R1335 is more flexible (*Figure 2C*) and weakens its binding to the G3 in favour of the adjacent backbone (*Figure 2D*). When xCas9 binds AAG, R1333 forms hydrogen bonds with the A1 backbone, and R1335 predominantly interacts with the G3 nucleobase. Consistent with the analysis in *Figure 2B*, for ignored PAM sequences, both arginines fail to interact with the PAM nucleotides, while displaying a substantial increase in interactions with non-PAM residues.

We then computed a specificity index by measuring the frequency of hydrogen bond formation between a given arginine and the PAM nucleotides relative to the frequency of forming hydrogen bonds with non-PAM residues (*Figure 2E*). Remarkably, R1335 exhibits a distinct specificity pattern for PAM sequences recognised by xCas9 compared to those that are ignored. This suggests that R1335 may serve as a discriminator for recognising specific PAM sequences in xCas9. In contrast, R1333, which possesses greater flexibility (*Figure 2C*), engages significantly with non-PAM residues (*Figure 2B and D*) rendering it non-specific (*Figure 2E*).

This demonstrates that in SpCas9, the arginine dyad is more rigid than in xCas9 (with R1335 more constrained than R1333). This rigidity enforces a strict selection of their preferred binding partner, which is guanine (*Hossain et al., 2023*; *Luscombe et al., 2001*). Contrarily, the E1219V substitution in xCas9 leads to increased flexibility of R1333/R1335, adjusting their interactions and expanding the number of PAM sequences that are recognised. However, to enable PAM recognition, effective interactions with the PAM nucleotides are required. Accordingly, the vast majority of recognised PAM sequences contain at least one guanine, prone to interact with arginine (*Hossain et al., 2023*; *Luscombe et al., 2001*). Ignored PAM sequences do not form any interaction between the PAM nucleotides and the arginine dyad, which shifts away from PAM (*Figure 2F*).

## TGG binding is entropically favoured in xCas9

To delve deeper into the role of the R1335, we performed free energy simulations. Well-tempered metadynamics simulations (*Barducci et al., 2008*) were carried out to elucidate the preference of R1335 to bind either the G3 nucleobase or the neighbouring E1219 in SpCas9 (*Figure 3A*, details in Materials and methods). Towards this aim, the free energy landscape was described along two collective variables (CVs) that define the distance of the R1335 guanidine from the E1219 carboxylic group (CV1) or the G3 nucleobase (CV2, details in Materials and methods), and sampled through µs-long converged well-tempered metadynamics simulations (*Figure 3—figure supplement 1*). A well-defined free energy minimum at ~0.4 nm indicates that R1335 stably binds and is effectively 'sandwiched' between E1219 and G3 (*Figure 3A*), which explains its rigidity and limited ability to recognise only NGG PAMs in SpCas9. We then sought to understand why xCas9 exhibits improved recognition of the TGG PAM sequence compared to SpCas9 (*Hu et al., 2018*). To investigate this, we conducted well-tempered metadynamics simulations focusing on the binding of R1335 to the G3 nucleobase and the DNA backbone in both SpCas9 and xCas9 (*Figure 3B and C*). The free energy was described along the distances between the R1335 guanidine and either the backbone phosphate group (CV1) or the G3 functional group atoms (CV2, details in Materials and methods). The obtained free energy landscapes reveal that in SpCas9, R1335 predominantly interacts with the G3 nucleobase (*Figure 3B*, *Figure 3—figure supplement 2*). This finding aligns with hydrogen bond analysis (*Figure 2D*) and is consistent with R1335 rigidity (*Figure 2C*). Such rigidity incurs an entropic cost due to the restricted conformational space of R1335 in the DNA-bound state. Conversely, in xCas9, R1335 exhibits dynamic interactions with the G3 nucleobase and backbone (*Figure 3B*, *Figure 3—figure supplement 3*). This increased flexibility reduces the entropic penalty for DNA binding, as revealed by entropy calculations

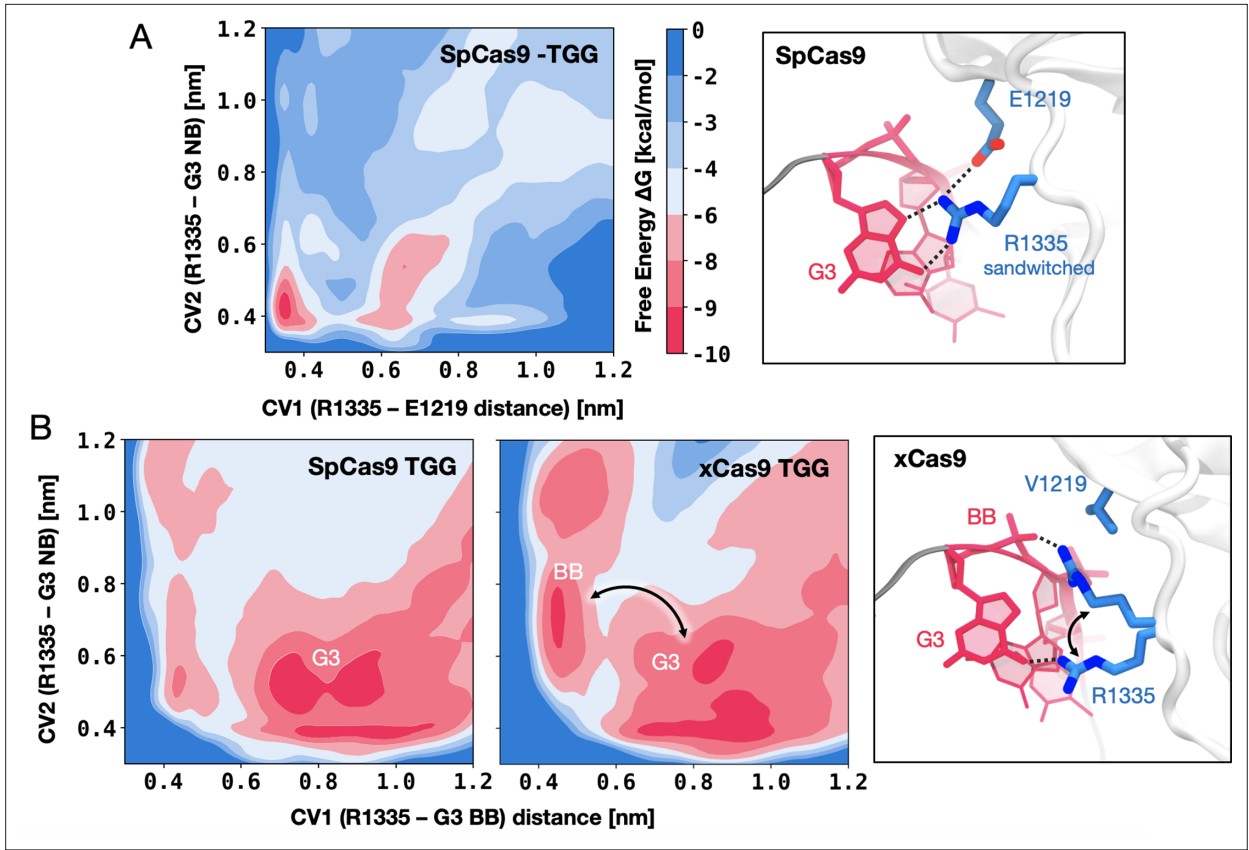

**Figure 3.** DNA-binding preference of R1335 in SpCas9 vs. xCas9. (**A**) Free energy surface (FES) describing the preference of R1335 for binding either the G3 nucleobase or E1219 in SpCas9. The FES is plotted along the distances between the R1335 guanidine and either the E1219 carboxylic group (CV1) or the G3 nucleobase (CV2, details in Materials and methods). A well-defined minimum indicates that R1335 is 'sandwiched' between G3 and E1219 (right). (**B**) FES describing the binding of R1335 to G3 and the DNA backbone in SpCas9 (left) and xCas9 (centre). The FES is plotted along the distances between the R1335 guanidine and either the backbone phosphate (CV1) or G3 nucleobase (CV2). In SpCas9, R1335 mainly binds the G3 nucleobase, while in xCas9, it alternates interactions between the nucleobase and backbone (right). The free energy, $\Delta G$, is expressed in kcal/mol (*Figure 3—figure supplements 1–3*).

The online version of this article includes the following figure supplement(s) for figure 3:

**Figure supplement 1.** Convergence of well-tempered metadynamics simulations characterising the preference of R1335 for binding either the G3 nucleobase or E1219 in SpCas9.

**Figure supplement 2.** Convergence of well-tempered metadynamics simulations characterising the binding of R1335 to G3 and the DNA backbone in SpCas9.

**Figure supplement 3.** Convergence of well-tempered metadynamics simulations characterising the binding of R1335 to G3 and the DNA backbone in xCas9.

using the quasi-harmonic approximation (details in Materials and methods). Specifically, R1335 in xCas9 exhibits an entropy increase of 216.83 J/mol·K compared to SpCas9 (320.01 J/mol·K for xCas9 vs. 103.18 J/mol·K for SpCas9). This reduction in entropic penalty allows for a more adaptable interaction between R1335 and the DNA, accommodating the DNA's inherent conformational flexibility. These findings provide a mechanistic explanation for xCas9's enhanced recognition of the TGG PAM compared to SpCas9 (*Hu et al., 2018*). In SpCas9 the interactions between the G3 nucleobase and R1335 result in an enthalpic gain. However, the rigidity of R1335 imposes a significant entropic cost. In contrast, the increased flexibility of R1335 in xCas9 minimises this entropic penalty, improving the recognition and binding of the TGG PAM.

## Directed evolution improves DNA binding

xCas9 was developed through three cycles of directed evolution. The E480K, E543D, and E1219V amino acid substitutions were introduced in the first evolution cycle (leading to xCas9$_1$); A262T, S409I,

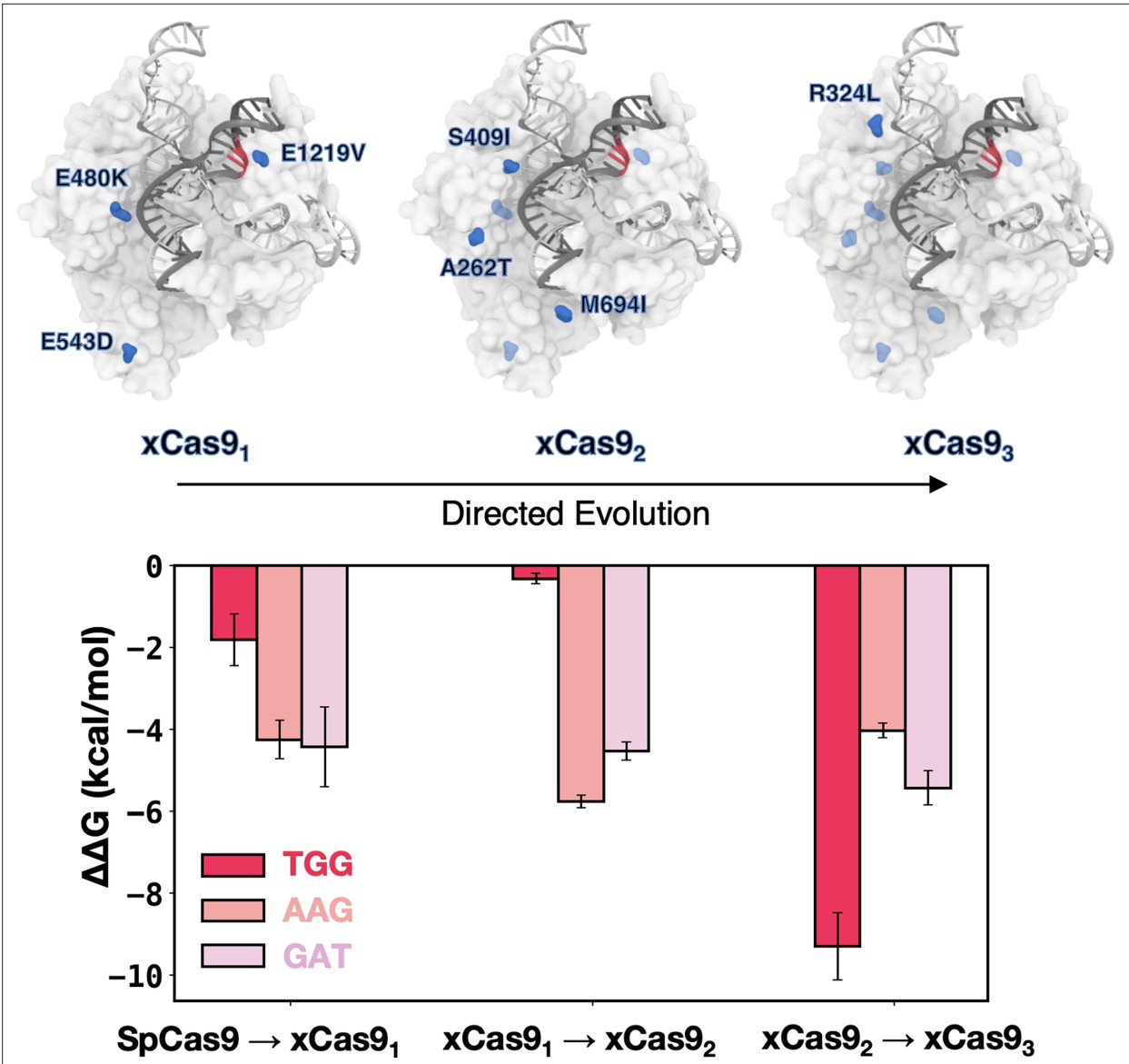

**Figure 4.** DNA-binding free energy difference (ΔΔG) between SpCas9 and its xCas9$_{1-3}$ mutants. Relative changes in the DNA-binding free energy (ΔΔG) upon transitioning from SpCas9 to xCas9$_1$, from xCas9$_1$ to xCas9$_2$, and from xCas9$_2$ to xCas9$_3$ in the presence of TGG (red), AAG (salmon), and GAT (pink) protospacer adjacent motif (PAM) sequences. Binding free energies from alchemical free energy calculations denoted with the associated error computed through the multistate Bennett acceptance ratio (MBAR) method (**Shirts and Chodera, 2008**) (details in Materials and methods) (**Figure 4—figure supplements 1 and 2**).

The online version of this article includes the following figure supplement(s) for figure 4:

**Figure supplement 1.** Thermodynamic cycle.

**Figure supplement 2.** Convergence of alchemical free energy calculations.

and M694I were introduced in the second evolution cycle (yielding xCas9$_2$); and R324L was included in the third evolution cycle (yielding xCas9$_3$) (**Hu et al., 2018**).

To assess how these substitutions contribute to expanding PAM recognition, we systematically introduced them into SpCas9 in a stepwise manner, following the evolution cycles, and evaluated their impact on DNA binding. We considered the TGG PAM sequence, effectively recognised by both SpCas9 and xCas9, as well as the AAG and GAT sequences, specifically recognised by xCas9. The contribution of each mutation cycle was computed as the difference in the DNA-binding free energy (ΔΔG) between SpCas9 and its mutant counterparts: xCas9$_1$, followed by xCas9$_2$, and finally xCas9$_3$

(*Figure 4*). We performed alchemical free energy calculations using a thermodynamic cycle (details in Materials and methods), obtaining $\Delta\Delta G$ values by transforming the respective amino acid residues in the presence or absence of bound DNA (*Figure 4—figure supplements 1 and 2*).

The transition from SpCas9 to xCas9$_1$ for TGG PAM does not significantly improve binding ($\Delta\Delta G$=−1.5 ± 0.63 kcal/mol), while the same transition for AAG and GAT PAMs is notably favoured ($\Delta\Delta G$=−4.25 ± 0.47 kcal/mol and −4.43±0.98 kcal/mol, respectively), consistent with xCas9's ability to recognise AAG and GAT. Upon introducing the second cycle of mutations (i.e. transforming xCas9$_1$ into xCas9$_2$), we observe a significant improvement in DNA binding in the presence of AAG and GAT ($\Delta\Delta G$=−5.76 ± 0.16 kcal/mol and −4.53±0.22 kcal/mol, respectively) while no substantial change is noted for TGG PAM (*Figure 3*). Hence, the recognition of non-canonical PAMs (i.e. AAG and GAT) improves in the second evolution cycle, as we further transition towards xCas9. It also shows that E1219V in the PAM-interacting domain is not the sole contributor to expanded PAM compatibility; and subsequent mutations play a crucial role. Finally, upon transitioning from xCas9$_2$ to xCas9$_3$, by introducing R324L, we observe a significant improvement in the DNA-binding free energy in the presence of TGG (by −9.29±0.82 kcal/mol), while the affinity for AAG and GAT only increased by −4.03±0.18 kcal/mol and 5.24±0.42 kcal/mol, respectively. This suggests that the third cycle of mutation is the key player behind the enhanced recognisability of the canonical TGG PAM by xCas9. This observation suggests that, while xCas9 expands DNA recognition towards non-canonical PAMs by improving DNA binding in the early evolution cycles, the enhanced recognisability of the canonical TGG PAM by xCas9 compared to SpCas9 observed experimentally primarily arises from the third cycle of mutations.

## Non-canonical PAMs induce a conformational change

To better understand the PAM-binding mechanism, we analysed the individual per-residue contributions to the DNA-binding free energy. Specifically, we examined how these contributions change during the transition from SpCas9 to xCas9$_1$, from xCas9$_1$ to xCas9$_2$, and from xCas9$_2$ to xCas9$_3$. These changes represent the difference in enthalpic contribution ($\Delta E$) to the DNA-binding free energy ($\Delta\Delta G$) (details in Materials and methods).

For the SpCas9 → xCas9$_1$ transition, we computed these contributions for the critical residue R1335 (adjacent to the E1219V) and the E480K and E543D substitutions (*Figure 5A*). We observe that R1335 interacts more favourably with the AAG- and GAT-bound xCas9$_1$ compared to TGG-bound xCas9$_1$, while the contribution of E480K is similar in all the studied systems. Interestingly, the contribution of E543D is unfavourable only in the presence of TGG, owing to a conformational change in the REC3 domain upon non-canonical PAMs binding. This finding aligns with the structural characterisations of xCas9. Superimposing xCas9 bound to TGG (PDB 6K4P) (*Chen et al., 2019*) onto the TGG-bound SpCas9 (PDB 4UN3) (*Anders et al., 2014*) reveals no significant differences, with a backbone RMSD of ~0.80 Å (*Figure 5—figure supplement 1*). In contrast, comparing TGG-bound SpCas9 with xCas9 bound to AAG and GAT (PDB 6AEB and 6AEG, respectively) (*Guo et al., 2019*) reveals a notable shift in the REC3 domain[7] (backbone RMSD of ~3.21 Å and~3.55 Å, respectively), consistent with our computational results. We also note that by computing the enthalpic contribution for the remaining protein residues, i.e., the $\Delta E$ as the overall interaction energy between the rest of the protein (without the selected residues) and the DNA, we observed no substantial difference for the TGG- and AAG-bound systems, while GAT is slightly more favourable (*Figure 5A*). Overall, this indicates that the contribution from the E543D substitution, being the only substitution of the first cycle located in REC3, is mainly influenced by the conformational change of REC3.

To further investigate whether the observed conformational change in the REC3 domain is due to the mutations incorporated in xCas9 or a result of binding alternative PAMs, we introduced the xCas9 mutations in the X-ray structure of SpCas9 (PDB 4UN3) and performed MD simulations in the presence of different PAM sequences, i.e., TGG, AAG, and GAT. These simulations were conducted in replicates, each reaching ~6 µs of sampling for each system, similar to our equilibrium MD simulations of xCas9 (details in Materials and methods).

To monitor the conformational change of REC3, we computed the centres of mass (COMs) distance between REC3 and HNH domains from the obtained MD trajectories (*Figure 5B*). The simulations reveal that in the presence of TGG, the REC3 domain maintains the configuration observed in the 4UN3 X-ray structure. On the other hand, in the presence of AAG and GAT, REC3 exhibits a notable opening of ~10–13 Å with respect to the 4UN3 X-ray structure, reaching a conformation that is similar

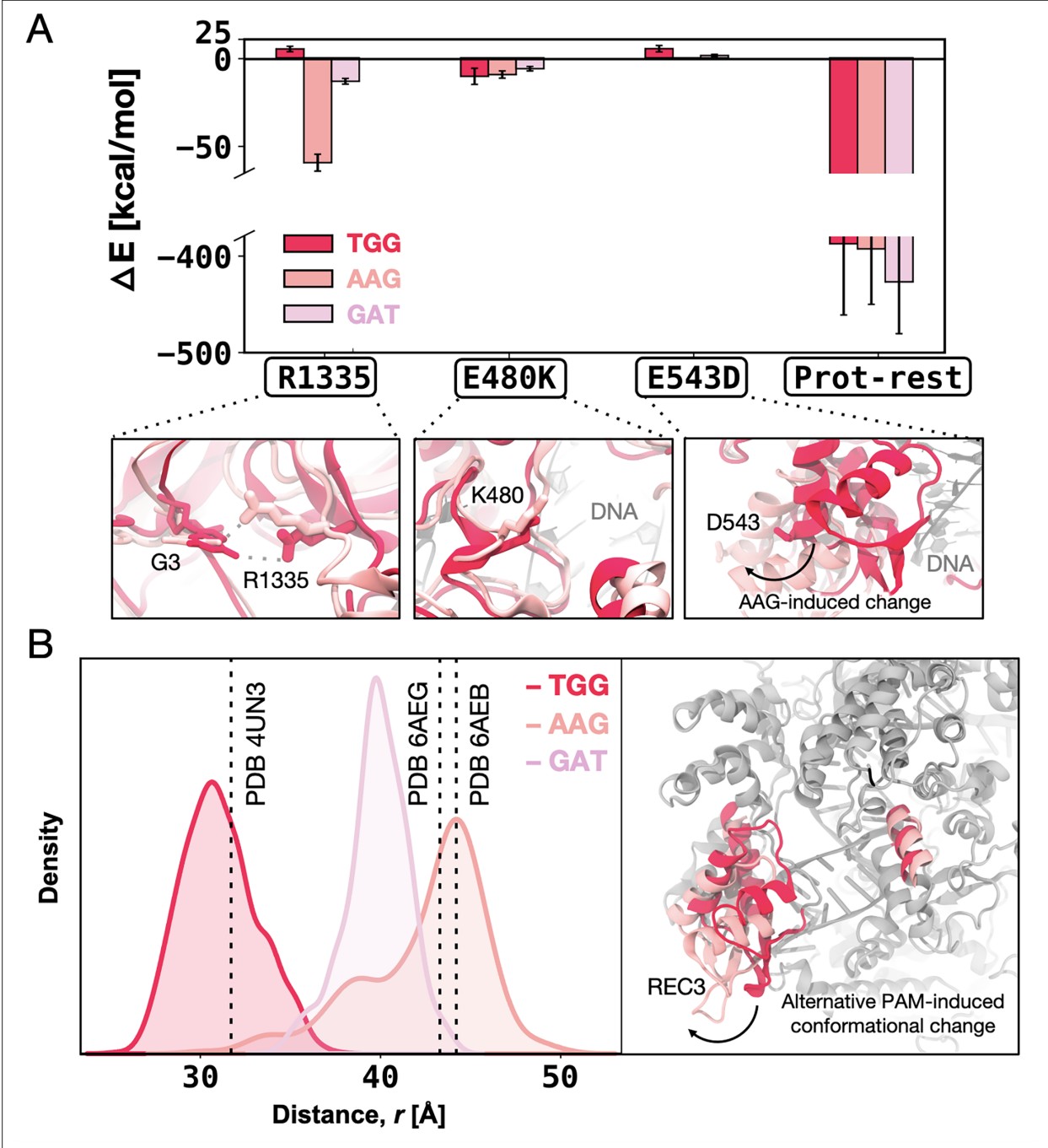

**Figure 5.** Enthalpic contribution to the DNA-binding free energy ($\Delta\Delta G$). (**A**) Enthalpic contribution to the $\Delta\Delta G$ of DNA binding while transitioning from SpCas9 to xCas9₁ in the presence of the TGG (red) and AAG (salmon), and GAT (pink) protospacer adjacent motif (PAM) sequences, computed as the average changes in the interaction energy ($\Delta E$) between selected amino acid residues and the DNA. The $\Delta E$ is also computed as the overall interaction energy between the rest of the protein (prot-rest, without the selected residues) and the DNA. Error bars were computed by averaging the results from different segments of the given trajectory (details in Materials and methods). (**B**) Probability density of the distance ($r$) between the centres of mass (COMs) of the REC3 and HNH domains from molecular dynamics simulations of SpCas9 (PDB 4UN3[1]) incorporating the xCas9 mutations in the presence of TGG (red) and AAG (pink) PAM sequences. Data from ~6 μs of aggregate sampling for each system. The values of the distance $r$ in the X-ray structures of SpCas9 (PDB 4UN3, 31.6 Å) and xCas9 bound to AAG (PDB 6AEB, 43.8 Å) and GAT (PDB 6AEG, 43.4 Å) are indicated using a vertical dashed bar. The statistical significance between the two distributions was evaluated using Z-score statistics with a two-tailed hypothesis (p-value was less than 0.0001). The distance ($r$) between the REC3 and HNH COMs is indicated on the three-dimensional structure of Cas9 (right), highlighting the AAG-induced conformational change using an arrow (*Figure 5—figure supplement 1*, *Figure 5—figure supplement 2*, and *Figure 5—figure supplement 3*).

*Figure 5 continued on next page*

**Figure supplement 1.** Comparison of existing structural data.

**Figure supplement 2.** Enthalpic contribution to the DNA-binding free energy for the $xCas9_1 \rightarrow xCas9_2$ transformation.

to their respective X-ray structures (PDB 6AEB and 6AEG). Together, our computations show that the REC3 conformational change is attributed to alternative PAM binding rather than the xCas9 amino acid substitutions, and occurs in the early cycle of directed evolution, suggesting that PAM binding acts as a positive allosteric effector of the REC3 function (*Nierzwicki et al., 2020*; *Palermo et al., 2017*; *Sternberg et al., 2015*; *Zuo and Liu, 2020*).

Analysis of the enthalpic contribution ($\Delta E$) to the $\Delta\Delta G$ upon the second evolution cycle (i.e. from $xCas9_1$ to $xCas9_2$) reveals that despite being distal from the DNA and PAM, the $xCas9_2$ mutations promote favourable interactions between R1335 and non-canonical PAM sequences (i.e. AAG and GAT with $\Delta E$ −26.92±6.33 and −38.60±7.18, respectively) (*Figure 5—figure supplement 2*). Upon the third evolution cycle (i.e. from $xCas9_2$ to $xCas9_3$), the enthalpic contribution to DNA binding does not report substantial differences for R1335 and R324L between TGG- and non-canonical PAMs-bound xCas9 (*Figure 5—figure supplement 3*). On the other hand, however, the overall interaction energy between the protein and the DNA is highly favourable for the TGG-bound $xCas9_3$, compared to the non-canonical PAMs-bound systems. This is notable considering that the overall $\Delta E$ for protein DNA binding between the TGG- and non-canonical PAMs-bound systems was negligible (slightly favourable for GAT) in the first cycle (*Figure 5A*), while more favourable for both AAG and GAT in the second cycle (*Figure 5—figure supplement 2*). This indicates that the third evolution cycle highly adapts to binding TGG, resulting in energetically favourable DNA binding compared to non-canonical PAMs. This observation contributes to clarifying how xCas9 expands PAM recognition while also improving recognition of the canonical TGG PAM compared to SpCas9.

## Discussion

Structural studies have shown that xCas9 can bind alternative PAM sequences, such as AAG and GAT (*Guo et al., 2019*), but they do not clarify how the enzyme simultaneously expands its DNA targeting capabilities while enhancing recognition of the canonical PAM. Here, we demonstrate that in SpCas9, the arginine dyad R1333 and R1335 within the PAM-interacting domain – particularly the rigidity of R1335 – enforces strict guanine selection, thereby restricting PAM compatibility to sequences closely aligned with the canonical NGG motif. This selectivity arises from a significant entropic penalty imposed by the restricted conformational space of R1335 in the DNA-bound state, which is compensated by enthalpic gains through specific interactions with guanine. As a result, SpCas9's rigid binding mechanism constrains its ability to recognise PAM sequences beyond the canonical TGG.

In contrast, xCas9 achieves remarkable flexibility in R1335 through the E1219V mutation. This mutation allows R1335 to sample a broader conformational space, enabling effective interactions with the PAM nucleobases and the DNA backbone (*Figure 2A*). Consequently, xCas9 can recognise alternative PAMs, such as AAG and GAT. Importantly, this flexibility reduces the entropic penalty associated with DNA binding. The increased adaptability of R1335 not only facilitates binding to non-canonical PAMs but also enhances recognition of the canonical TGG PAM, as demonstrated through free energy simulations (*Figure 3*). This explains xCas9's enhanced recognition of the TGG PAM compared to SpCas9 (*Hu et al., 2018*). It is also notable that for the successful recognition of alternative PAMs effective interactions with the PAM nucleotides are required, underscoring a fundamental principle of molecular recognition: the balance between enthalpy and entropy. Accordingly, most recognised PAM sequences contain at least one guanine, which readily forms specific interactions with arginine (*Hossain et al., 2023*; *Luscombe et al., 2001*). On the other hand, for ignored PAM sequences (those lacking guanine), the arginine dyad shifts away from the PAM nucleotides, demonstrating selective targeting (*Figure 2F*).

xCas9 was developed through three cycles of directed evolution, progressively introducing a total of seven amino acid substitutions with respect to SpCas9. We evaluated the contribution of these

substitutions to expanded PAM recognition by incorporating them into SpCas9 in a stepwise fashion and assessed their effects on DNA binding. We found that the directed evolution process tunes the specificity of xCas9 for non-canonical PAM sequences by progressively enhancing the DNA-binding affinity. Specifically, the substitutions incorporated during the first evolution cycle primarily improved DNA binding with non-canonical PAMs, as evidenced by the substantial increase in binding free energy for AAG (*Figure 4*). Subsequent mutations in the second evolution cycle further stabilised these inter-actions, particularly with R1335, while the final mutation cycle significantly enhanced the binding affinity for the canonical TGG PAM, achieving an energetically favourable DNA-binding configuration. These findings indicate that the expanded DNA targeting capability is established early in the directed evolution process, while the enhanced recognition of the canonical TGG PAM by xCas9, as compared to SpCas9, predominantly arises from mutations introduced during the third cycle. These findings are particularly significant in the context of the extensive research on directed evolution. In our recent study on innovative CRISPR-Cas9-conjugated adenine base editors (ABEs), we have demonstrated that the exceptional efficiency of ABE8e, optimised through directed evolution, is driven by a dynamic interplay of destabilising and stabilising mutations (*Arantes et al., 2024*). Initial destabilising mutations enhanced base editing activity, while subsequent stabilising mutations strengthened Cas9 binding, collectively enabling the remarkable efficiency observed in ABE8e. Studies on directed evolution by *Romero and Arnold, 2009*, *Tokuriki et al., 2008*, and *Wang et al., 2002* have also shown that early substitutions in the evolution process commonly reshape functionality, while subsequent muta-tions confer a stabilising role. This pattern is evident in the enhanced DNA-binding affinity of xCas9 for the AAG and GAT PAMs, which was achieved in the initial evolution cycle. Conversely, structural mutations that stabilise the complex tend to occur in later stages, as exemplified by the substantial improvement in DNA-binding affinity for the canonical TGG PAM-bound xCas9 observed during the third cycle of evolution. This stepwise improvement underscores the cumulative effect of directed evolution in expanding PAM recognition while also fine-tuning the recognition of the canonical PAM sequence.

The recognition of alternative PAMs also triggers a conformational change in the REC3 domain (*Figure 5*), driven by alternative PAMs binding rather than the amino acid substitutions in xCas9. This conformational shift is observed upon AAG and GAT binding in the initial cycle of directed evolution and is maintained throughout the evolution of xCas9. The REC3 domain eventually adopts an open conformation similar to that seen in their respective X-ray structures (PDB 6AEB and 6AEG) (*Guo et al., 2019*). This observation suggests that the binding of alternative PAM sequences modifies the conformational dynamics of the distally located REC3 domain, implying an allosteric regulatory mech-anism. This aligns with the concept of Cas9 functioning as an allosteric engine, where the binding of the canonical PAM sequence primes the protein for double-stranded DNA cleavage (*Nierzwicki et al., 2020*; *Palermo et al., 2017*; *Sternberg et al., 2015*; *Zuo and Liu, 2020*). On the other hand, when bound to TGG, xCas9 retains a closed conformation of the REC3 domain, akin to that observed in the X-ray structure of the TGG-bound SpCas9 (PDB 4UN3) (*Anders et al., 2014*). As the third round of evolution enhances DNA binding in the presence of TGG (*Figure 4*), a compact and energetically favourable conformation is essential for the efficient recognition of the canonical TGG PAM.

The impact of alternative PAM sequences on the conformational changes in the REC3 dynamics (*Figure 5B*) highlights the need for novel studies to accurately elucidate the allosteric relationship between these distally located elements (*East et al., 2020*; *Nierzwicki et al., 2021*; *Skeens et al., 2024*). Indeed. the conformational change described here raises a series of intriguing questions, including the molecular triggers involved and the specific role of each mutation introduced at various stages. Most notably, the causal relationship between alternative PAM binding and the REC3 confor-mational change is an overarching question (*Hibshman et al., 2024*). To elucidate the molecular determinants driving these cause-effect relationships and the domino of events potentially associated with them, novel theoretical methods for inferring causality will need to be developed.

## Conclusions

This study identified the molecular determinants underlying xCas9 expanded DNA recognition through comprehensive molecular and free energy simulations. We demonstrate that the flexibility of the PAM-interacting residues, particularly R1335, is a key factor driving xCas9's ability to recognise a broader range of PAM sequences, compared to the *S. pyogenes* Cas9 (SpCas9). This flexibility reduces

the entropic penalty during DNA binding, broadening xCas9's functional range towards non-canonical PAMs, while maintaining a strong preference for guanine-containing motifs. This favourable binding in xCas9 reflects a broader evolutionary strategy in protein-DNA recognition, where structural flexibility enables sequence diversity without sacrificing specificity (*Chiu et al., 2022*; *Heller et al., 2020*; *Luscombe et al., 2001*; *Luscombe and Thornton, 2002*). xCas9 thereby exemplifies how flexibility-driven optimisation can expand protein functionality, offering a promising framework for engineering next-generation genome editing tools. In this context, introducing further substitutions within the PAM-interacting cleft may fine-tune the flexibility of the PAM-interacting residues, enhancing the recognition of adenine- and thymine-containing PAMs. This mechanism might be exploited by the PAM-less SpRY Cas9 variant, which is capable of recognising a wider array of PAMs (*Walton et al., 2020*). The mutations within its PAM-binding cleft likely increase the flexibility of the interacting residues (*Hibshman et al., 2024*), enabling broader PAM recognition and reduced sequence specificity. Hence, strategically targeting the flexibility of the PAM-interacting cleft improves enhanced adaptability and tailored specificities. Furthermore, as balancing flexibility is a conserved evolutionary strategy in protein-DNA recognition, our findings extend beyond the SpCas9 protein and may apply to other CRISPR-Cas effectors from diverse species, each with distinct PAM recognition requirements.

It is also notable that xCas9 expands the DNA targeting capability in the early cycles of directed evolution, while the enhanced recognition of the canonical TGG PAM by xCas9, as compared to SpCas9, predominantly arises from mutations introduced during the third cycle. These findings, combined with the entropically favourable interactions within the PAM-interacting cleft, elucidate how xCas9 broadens its PAM recognition repertoire while simultaneously improving its binding affinity for TGG (*Hu et al., 2018*). Finally, our simulations reveal an intriguing allosteric phenomenon in which the binding of alternative PAMs – such as AAG and GAT, but not TGG – induces a conformational change in the distally located REC3 domain. Experimental studies and further computational analysis will be essential to elucidate the causative mechanisms underlying this phenomenon and to clarify the specific roles of point mutations in driving these conformational changes.

Building on the insights from this study, future engineering should fine-tune the flexibility of the PAM-interacting cleft to design CRISPR-Cas systems with expanded genome targeting capabilities and tailored specificities. Such advancements could significantly enhance the versatility and precision of genome editing applications, from basic research to therapeutic interventions.

## Materials and methods
### Structural models
Molecular simulations have been based on the *S. pyogenes* Cas9 (SpCas9) and its variant xCas9 3.7 (hereafter referred to as xCas9) bound to several PAM sequences. The wild-type SpCas9 bound to the 5'-TGG-3' canonical PAM has been based on the X-ray crystallographic structure (PDB 4UN3) (*Anders et al., 2014*), solved at 2.59 Å resolution. Three systems of xCas9 bound to PAM sequences that are recognised were based on the X-ray structures of xCas9 including 5'-TGG-3' (PDB 6K4P [*Chen et al., 2019*], solved at 2.90 Å resolution), 5'-GAT-3' (PDB 6AEG [*Guo et al., 2019*], solved at 2.70 Å resolution), and 5'-AAG-3' (PDB 6AEB solved at 3.00 Å resolution [*Guo et al., 2019*]). xCas9 was also simulated bound to PAM sequences that are not recognised (5'-ATC-3', 5'-TTA-3', and 5'-CCT-3'). These systems were based on the PDB 6AEB (*Guo et al., 2019*). Three additional simulation systems were considered to examine the effect of alternative PAM binding on the conformational dynamics of Cas9. In detail, starting from the X-ray structure of SpCas9 (PDB 4UN3 [*Anders et al., 2014*]), xCas9 mutations were introduced in the presence of both TGG, AAG, and GAT PAM sequences. All simulation systems have been embedded in explicit waters, adding Na$^+$ and Cl$^-$ counterions to provide physiological ionic strength (0.15 M), and reaching ~340,000 atoms each (periodic box ~148.5 × 185.0 × 125.1 Å$^3$).

### MD simulations
MD simulations were performed through a simulation protocol tailored for protein/nucleic acid complexes (*Sinha et al., 2023*), which we also employed in studies of genome editing systems (*Pacesa et al., 2022*; *Saha et al., 2024*; *Sinha et al., 2024*). We used the Amber ff19SB force field (*Tian et al., 2020*), incorporating the OL15 (*Galindo-Murillo et al., 2016*) corrections for DNA and the

OL3 (*Banáš et al., 2010*; *Zgarbová et al., 2011*) corrections for RNA. The TIP3P model was employed for explicit water molecules (*Jorgensen et al., 1983*). The Li & Merz 12-6-4 model was used for Mg²⁺ ions (*Li and Merz, 2014*). All simulations were performed using the Gromacs 2018.5 code (*Abraham et al., 2015*). The simulations were conducted in the NPT ensemble using the leap-frog algorithm for integrating the equations of motion, with a time step of 2 fs. The temperature was maintained at 310 K using the v-rescale thermostat (*Bussi et al., 2007*) with a time constant of 0.1 ps. The pressure was controlled at 1 bar with the isotropic Parrinello-Rahman barostat (*Parrinello and Rahman, 1981*). Periodic boundary conditions were applied in three dimensions. Long-range electrostatic interactions were computed using the particle mesh Ewald approach (*York et al., 1993*) with a real-space cut-off of 1.2 nm and a Fourier grid spacing of 0.12 nm. Van der Waals interactions were modelled using the Lennard-Jones potential with a cut-off of 1.2 nm and a switching distance of 1 nm. The P-LINCS (*Hess, 2008*) algorithm was used to constrain bond lengths of the protein, DNA and RNA, and the SETTLE (*Miyamoto and Kollman, 1992*) algorithm was used to preserve the water molecules' geometry. Production runs were carried out collecting ~1.5 µs for each system and in four replicates. This resulted in a collective ensemble of ~6 µs for each system, totalling ~60 µs of simulated runs (i.e. ~6 µs for 10 simulation systems).

## Well-tempered metadynamics

To explore the molecular energetics underlying the interactions of R1335 and exhaustively sample the conformational space, we conducted two-dimensional well-tempered metadynamics simulations. Metadynamics is a non-equilibrium simulation method enabling the exploration of higher-dimensional free energy surfaces by reconstructing the probability distribution as a function of a few predefined CVs (*Bussi and Laio, 2020*; *Laio and Parrinello, 2002*). In metadynamics, the system's evolution is biased by a history-dependent potential, constructed through the cumulative addition of Gaussian functions deposited along the trajectory in the CVs space. As this bias potential compensates for the underlying free energy surface, the latter can be computed as a function of the CVs. Here, we performed well-tempered metadynamics (*Barducci et al., 2008*), an improvement that enhances the CV-space exploration by introducing a tuneable parameter $\Delta T$ that regulates the bias potential. We further improved the sampling through multiple walkers (*Minoukadeh et al., 2010*), to ensure efficient scan across the specified reaction CVs. In well-tempered metadynamics, the bias deposition rate decreases over the course of the simulation, which is achieved by using a modified expression for the bias potential, $V(s, t)$:

$$V(s, t) = \sum_{t'=0, t_G, 2t_G, \dots}^{t' < t} \omega t_G e^{-V\left(s\left(q\left(t'\right), t'\right)\right)/\Delta T} e^{-\sum_{i=1}^{2}\left[\left(s_i(q) - s_i\left(q\left(t'\right)\right)\right)^2/2\sigma_i^2\right]} \tag{1}$$

where $\omega$ is the deposition rate and $t_G$ is the deposition stride of the Gaussian hills. Here, each simulation applied a biasing potential with the deposition rate ($\omega$) and deposition stride ($t_G$) of the Gaussian hills of 0.239 kcal/mol/ps (1.0 kJ/mol/ps) and 10 ps, respectively. The bias factor $(T + \Delta T)/T$ was set to 4 at 300 K and the free energy, $F(s, t)$, was then computed as:

$$F(s, t) = -\frac{T + \Delta T}{\Delta T}\left(V(s, t) - C(T)\right) \tag{2}$$

where $\Delta T$ is the difference between the temperature of the CV and the simulation temperature $T$. The bias potential is grown as the sum of the Gaussian hills deposited along the CV space, with the sampling of the CV space being controlled by the tuneable parameter $\Delta T$. All well-tempered metadynamics simulations were started with a well-equilibrated structure generated from unbiased MD simulation.

Well-tempered metadynamics was carried out using two CVs for each simulation. To examine the interactions between R1335 and either E1219 or the G3 nucleobase in SpCas9 (*Figure 3A*), the free energy landscape was studied along the distances between the COMs of the R1335 guanidinium group and either the carboxylic group of E1219 (CV1) or the G3 functional group atoms exposed in the major groove (O6 and N7; CV2). To compare the interactions of R1335 in SpCas9 and xCas9 (*Figure 3B and C*, main text) and evaluate its preference for binding either the G3 nucleobase or the DNA backbone, the CVs were defined as the COM distances between the R1335 guanidinium group

and either the backbone phosphate group atoms (OP1, OP2, P; CV1) or the COM of the G3 functional group atoms (O6 and N7; CV2). To restrict the sampled range of coordinates, one-sided harmonic potentials with a force constant of 3500 kJ/mol·nm$^2$ were employed, limiting the range of the CVs to 0.3–1.2 nm. Each well-tempered metadynamics simulation was carried out for ~1 μs, reaching converged free energy surfaces (*Figure 3—figure supplements 1–3*). Well-tempered metadynamics simulations were performed using the Gromacs 2018.5 code (*Abraham et al., 2015*) and the open-source, community-developed PLUMED library (*Tribello et al., 2014*).

## Alchemical free energy calculations

Alchemical free energy calculations were performed to assess the impact of mutations introduced into SpCas9 during directed evolution and resulting in xCas9, on the DNA-binding affinity. We determined the binding free energy difference ($\Delta\Delta G$) between SpCas9 and its xCas9 mutants in the presence of the TGG, AAG, and GAT PAM sequences. We considered three xCas9 mutants: $xCas9_1$, $xCas9_2$, and $xCas9_3$, which emerged through three successive cycles of directed evolution (*Hu et al., 2018*). In detail, starting from the X-ray structure of SpCas9 (PDB 4UN3)[1], the E480K, E543D, and E1219 mutations were introduced during the first evolution cycle (yielding $xCas9_1$); A262T, S409I, and M694I arose in the second cycle ($xCas9_2$); and R324L was introduced in the third cycle ($xCas9_3$). The relative $\Delta\Delta G$ of binding was thereby computed while moving from SpCas9 to $xCas9_1$, from $xCas9_1$ to $xCas9_2$, and from $xCas9_2$ to $xCas9_3$. This approach involved defining two end states, commonly referred to as 'state A' ($\lambda = 0$; e.g. SpCas9) and 'state B' ($\lambda = 1$; e.g. $xCas9_1$), using different molecular topologies to represent the initial and final states of a chemical process. We utilised a thermodynamic cycle (*Figure 4—figure supplement 1*) enabling us to compute the free energies associated with the 'alchemical' transformation of specific amino acid residues in the absence ($\Delta G_{m1}$) or presence ($\Delta G_{m2}$) of the DNA substrate bound to the respective Cas9. This transformation was achieved by simulating the system independently for various values of the scaling parameter $\lambda$ (referred to as $\lambda$ windows) ranging from 0 to 1, to linearly interpolate between the potential energy functions of the physical end states. Due to the need for multiple amino acid mutations in each transition (e.g. from SpCas9 to xCas9), a large number of $\lambda$-windows were required, significantly increasing the computational cost. In detail, the distribution of the $\lambda$ windows were optimised using a gradient descent algorithm to maximise the probabilities of exchange between adjacent states (https://gitlab.com/KomBioMol/converge_lambdas; *KomBioMol, 2021*; *Wieczor and Czub, 2022*). The neighbouring $\lambda$ windows were allowed to exchange their configurations every 0.5 ps according to the Metropolis criterion, and the values of $\lambda$ were optimised to achieve the acceptance rate of at least 10%. Since each intermediate $\lambda$ state is technically a hybrid between the $A$ and $B$ end point states, we generated their dual coordinates and topologies using the PMX server (http://pmx.mpibpc.mpg.de/) (*Gapsys and de Groot, 2017*). The relative free energy changes (i.e. $\Delta\Delta G$) for the transformation were computed using the multistate Bennett acceptance ratio method (*Matsunaga et al., 2022*; *Shirts and Chodera, 2008*) to integrate the free energies over the different $\lambda$ values (*Klimovich et al., 2015*). Each system underwent simulation for a minimum of ~80 ns in each $\lambda$ window, collecting ~18 μs of simulated runs, until reasonable convergence of $\Delta\Delta G$ was attained (*Figure 4—figure supplement 2*). Hamiltonian-replica exchange (*Hritz and Oostenbrink, 2008*) was used in each simulation, with exchanges attempted every 1000 steps (or 2 ps), to enhance the sampling efficiency and ensure adequate overlap between neighbouring windows. The enthalpic contributions to the DNA-binding free energy ($\Delta\Delta G$) were computed as the average changes in the interaction energy ($\Delta E$) between selected amino acid residues and the DNA. In detail, the $\Delta E$ values were calculated from the FEP trajectories, considering only the physical states (i.e. $\lambda = 0$ and $\lambda = 1$) and discarding the first ~10% frames as the equilibration phase. The values presented in *Figure 5A*, *Figure 5—figure supplement 2*, and *Figure 5—figure supplement 3* represent the average over the trajectory. The error estimation of the average was determined based on block averages over five blocks using the Gromacs 2018.5 energy module (*Abraham et al., 2015*).

## Analysis of structural data

The probability for R1333 and R1335 to interact with the PAM nucleobases (PAM NB), the PAM backbone (PAM BB), and non-PAM nucleotides (non-PAM) (*Figure 2b*) was computed as follows. The contacts between two arginine residues (R1333 and R1335) and the DNA duplex were defined based on the two criteria: (1) the distance between the COM of the arginine guanidinium group and

either of the COM of the DNA bases' heavy atoms or the COM of the backbone phosphate groups and (2) the interaction energies between the arginine residues and the DNA bases or the backbone phosphate groups. As possible contacts, each base-guanidine and phosphate-guanidine pairs were selected for which the COM distance was below 0.6 nm and 0.5 nm, respectively, in at least 5% of the cumulative trajectories. For these pairs, interaction energies between arginine residues and either the DNA bases or the backbone phosphates were computed. As a criterion to define effective contacts, the interaction strength of at least 100 kJ/mol and 350 kJ/mol was used for the arginine-base and arginine-phosphate pairs, respectively. These criteria allowed distinguishing the pairs that form efficient interactions from the pairs that are close to each other, but do not properly interact (such as contacts between the arginine guanidinium groups and bases adjacent to the properly interacting nucleotide). The interaction frequencies (*Figure 2B*) were computed through the following process. Initially, we classified whether a given interaction occurred using predefined energy and distance criteria (as described above). This classification yielded binary data, which we treated as a Bernoulli distribution to compute the variance of the interaction frequencies. Next, to estimate the number of independent data points, we calculated the autocorrelation of the interaction energy data. For each interaction, we utilised the largest autocorrelation time derived from four independent simulation replicates (of ~1.5 μs each, totalling ~6 μs per system) to determine the number of effective independent samples. Finally, we used the computed variance and number of independent samples to compute the error of each interaction mean. This methodology ensured a robust estimation of the mean interaction frequency and its associated error.

Hydrogen bonds between the arginine side chains and the PAM NB, BB, and non-PAM nucleotides were analysed using the Gromacs 2018.5 *hbond* analysis tool (*Abraham et al., 2015*). The standard geometrical criteria applied were a cut-off value of 3.5 Å for the acceptor-donor distance and 30° for the hydrogen-donor-acceptor angle. Hydrogen bond frequencies (*Figure 2D*) were calculated through a binary classification of the presence or absence of hydrogen bonds. The mean frequency was determined as the ratio of the number of hydrogen bonds in a given trajectory to the total number of frames. Data normalisation was achieved using a standard method, which involved dividing by the sum of all elements in the given dataset (e.g. hydrogen bonds for R1333 with PAM-NB, PAM-BB, and non-PAM nucleotides). This normalisation ensured comparability across different interaction types or studied systems. These calculations were performed considering the overall ensemble of ~6 μs for each system (*Figure 2D*), as well as for the independent simulation replicates of ~1.5 μs each (*Figure 2—figure supplement 1*).

The conformational entropy of the R1335 residue in SpCas9 and xCas9 bound to TGG PAMs was calculated using the quasi-harmonic approximation (*Karplus and Kushick, 1981*). This analysis was performed on well-equilibrated MD trajectories for each system. The covariance matrix of atomic fluctuations was constructed with the *gmx covar* tool in Gromacs (*Abraham et al., 2015*), focusing on the R1335 residue after removing its rotational and translational motions to ensure an appropriate representation of internal conformational fluctuations. Eigenvectors and eigenvalues were subsequently extracted using the *gmx anaeig* tool, and entropy was computed from the eigenvalues, which are representative of the vibrational modes of the system.

## Acknowledgements

We thank Dr. Pablo R Arantes for useful discussions. This material is based upon work supported by the National Institutes of Health (Grant No. R01GM141329 to GP) and the National Science Foundation (Grant No. CHE-2144823 to GP). GP acknowledges support by the Alfred P Sloan Foundation (Grant No. FG-2023-20431) and the Camille and Henry Dreyfus Foundation (Grant No. TC-24-063). KAH and MO acknowledge support by EU-HPC, BioExcel 101093290, Horizon-EU MDDB 101094561, and Spanish MCIN PDI2021-122478NB-I00. This work used Expanse at the San Diego Supercomputing Center through allocation MCB160059 and Bridges2 at the Pittsburgh Supercomputer Center through allocation BIO230007 from the Advanced Cyberinfrastructure Coordination Ecosystem: Services & Support (ACCESS) program, which is supported by National Science Foundation supports, grants #2138259, #2138286, #2138307, #2137603, and #2138296.

## Additional information

### Funding

| Funder | Grant reference number | Author |
|---|---|---|
| National Science Foundation | CHE-2144823 | Giulia Palermo |
| Alfred P. Sloan Foundation | FG-2023-20431 | Giulia Palermo |
| Camille and Henry Dreyfus Foundation | TC-24-063 | Giulia Palermo |
| HORIZON EUROPE Framework Programme | 101094561 | Kazi A Hossain Modesto Orozco |
| EU-HPC, BioExcel | 10.3030/101093290 | Kazi A Hossain Modesto Orozco |
| Spanish Ministry of Science and Innovation | PDI2021-122478NB-I00 | Modesto Orozco |
| National Institutes of Health | R01GM141329 | Giulia Palermo |

The funders had no role in study design, data collection and interpretation, or the decision to submit the work for publication.

### Author contributions

Kazi A Hossain, Conceptualization, Data curation, Formal analysis, Investigation, Visualization, Methodology, Writing – original draft; Lukasz Nierzwicki, Conceptualization, Data curation, Formal analysis, Investigation, Methodology, Writing – review and editing; Modesto Orozco, Jacek Czub, Resources, Supervision, Writing – review and editing; Giulia Palermo, Conceptualization, Resources, Supervision, Funding acquisition, Validation, Visualization, Project administration, Writing – review and editing

### Author ORCIDs

Kazi A Hossain ⓘ https://orcid.org/0000-0002-1149-964X
Modesto Orozco ⓘ https://orcid.org/0000-0002-8608-3278
Giulia Palermo ⓘ https://orcid.org/0000-0003-1404-8737

Joint Public Review: https://doi.org/10.7554/eLife.102538.3.sa1
Author response https://doi.org/10.7554/eLife.102538.3.sa2

## Additional files

### Supplementary files

MDAR checklist

### Data availability

The files from the molecular dynamics simulations, including trajectory and visualization data, and processed data (for Figures 2 to 5) can be accessed on Dryad.

The following dataset was generated:

| Author(s) | Year | Dataset title | Dataset URL | Database and Identifier |
|---|---|---|---|---|
| Hossain KA, Nierzwicki L, Orozco M, Czub J, Palermo G | 2025 | Data from: Mechanism of expanded DNA recognition in xCas9 | https://doi.org/10.5061/dryad.0000000dt | Dryad Digital Repository, 10.5061/dryad.0000000dt |

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
