## [Editor Report · eLife Assessment]

This manuscript describes a **fundamental** investigation of the functioning of Cas9 and in particular on how variant xCas9 expands DNA targeting ability by an increase-flexibility mechanism. The authors provide **compelling** evidence to support their mechanistic models and the relevance of flexibility and entropy in recognition. This work can be of interest to a broad community of structural biophysicists, computational biologists, chemists, and biochemists.

---

## [Referee Report · Joint Public Review]

Summary:

Hossain and coworkers investigate the mechanisms of recognition of xCas9, a variant of Cas9 with expanded targeting capability for DNA. They do so by using molecular simulations and combining different flavors of simulation techniques, ranging from long classical MD simulations, to enhanced sampling, to free energy calculations of affinity differences. Through this, the authors are able to develop a consistent model of expanded recognition based on the enhanced flexibility of the protein receptor.

Strengths:

The paper is solidly based on the ability of the authors to master molecular simulations of highly complex systems. In my opinion, this paper shows no major weaknesses. The simulations are carried out in a technically sound way. Comparative analyses of different systems provide valuable insights, even within the well-known limitations of MD. Plus, the authors further investigate why xCas9 exhibits improved recognition of the TGG PAM sequence compared to SpCas9 via well-tempered metadynamics simulations focusing on the binding of R1335 to the G3 nucleobase and the DNA backbone in both SpCas9 and xCas9. In this context, the authors provide a free-energy profiling that helps support their final model.

The implementation of FEP calculations to mimic directed evolution improvement of DNA binding is also interesting, original and well-conducted.

Overall, my assessment of this paper is that it represents a strong manuscript, competently designed and conducted, and highly valuable from a technical point of view.

Weaknesses:

To make their impact even more general, the authors may consider expanding their discussion on entropic binding to other recent cases that have been presented in the literature recently (such as e.g. the identification of small molecules for Abeta peptides, or the identification of "fuzzy" mechanisms of binding to protein HMGB1). The point on flexibility helping adaptability and expansion of functional properties is important, and should probably be given more evidence and more direct links with a wider picture.

Comments on revisions:

We have read the revised version and the response letter and I find that this manuscript is ready. There is no need for further additions/revisions.

---

## [Author Response]

The following is the authors’ response to the original reviews.

**Joint Public Review:**
Strengths:The paper is solidly based on the ability of the authors to master molecular simulations of highly complex systems. In my opinion, this paper shows no major weaknesses. The simulations are carried out in a technically sound way. Comparative analyses of different systems provide valuable insights, even within the well-known limitations of MD. Plus, the authors further investigate why xCas9 exhibits improved recognition of the TGG PAM sequence compared to SpCas9 via well-tempered metadynamics simulations focusing on the binding of R1335 to the G3 nucleobase and the DNA backbone in both SpCas9 and xCas9. In this context, the authors provide a free-energy profiling that helps support their final model.The implementation of FEP calculations to mimic directed evolution improvement of DNA binding is also interesting, original and well-conducted.

We thank the reviewer for their positive evaluation of our computational strategy. To further substantiate our findings, we have incorporated additional molecular dynamics and Free Energy Perturbation (FEP) calculations for the system bound to GAT. These results corroborate our previous observations obtained with AAG, reinforcing our conclusions.

Overall, my assessment of this paper is that it represents a strong manuscript, competently designed and conducted, and highly valuable from a technical point of view.Weaknesses:To make their impact even more general, the authors may consider expanding their discussion on entropic binding to other recent cases that have been presented in the literature recently (such as e.g. the identification of small molecules for Abeta peptides, or the identification of "fuzzy" mechanisms of binding to protein HMGB1). The point on flexibility helping adaptability and expansion of functional properties is important, and should probably be given more evidence and more direct links with a wider picture.

We have expanded our discussion on the role of entropy in favoring TGG binding to xCas9. To this end, we performed entropy calculations using the Quasi-Harmonic approximation (details provided in the Materials and Methods section). This analysis reveals that R1335 in xCas9 experiences an entropy increase compared to SpCas9, enhancing its adaptability and interaction with the DNA. This analysis and its explanation are detailed on pages 8-9.

Additionally, we have enriched the Discussion section by clarifying how DNA binding is entropically favored in xCas9, thereby facilitating the recognition of alternative PAM sequences. A refined explanation is also included in the Conclusions section, where we contextualize xCas9 within a broader evolutionary framework of protein-DNA recognition. This highlights how structural flexibility can enable sequence diversity while maintaining high specificity.

**Recommendations for the authors:**
Overall, this is a very interesting and elegant manuscript with compelling results that shed light on the atomistic determinants of genetic-editing technologies.Since the paper proposes new findings that may be helpful for experimentalists, it would be interesting if the authors point out (in their discussion/conclusions) specific amino acids to mutate/target for future tests by the experimental community. This should just appear as an open hypothesis/proposal for new experiments.

In the Conclusions, we have incorporated a discussion on how modifications in the PAM-binding cleft can enhance the recognition of alternative PAM sequences. As an illustrative example, we reference the recently developed SpRY Cas9 variant, which is capable of recognizing a broader range of PAMs. This variant includes mutations within the PAM-binding cleft that likely increase the flexibility of the interacting residues, as suggested by recent cryo-EM structures (Hibshman et al. Nat. Commun. 2024). The importance of fine-tuning the flexibility of the PAM-interacting cleft for engineering strategies has also been highlighted in the abstract.

Overall, in light of the reviewer’s comments and in consideration of our findings, we revised the manuscript title in: “Flexibility in PAM Recognition Expands DNA Targeting in xCas9.” This new title better highlights the key findings from our research and contextualizes them within the broader goal of expanding DNA targeting capabilities, a critical priority for developing enhanced CRISPR-Cas systems.